# Crotamine/siRNA Nanocomplexes for Functional Downregulation of Syndecan-1 in Renal Proximal Tubular Epithelial Cells

**DOI:** 10.3390/pharmaceutics15061576

**Published:** 2023-05-23

**Authors:** Joana D’Arc Campeiro, Wendy A. Dam, Mirian A. F. Hayashi, Jacob van den Born

**Affiliations:** 1Department Nephrology, University Medical Center Groningen, University of Groningen, Hanzeplein 1, De Brug, 4th Floor, AA53, 9713 GZ Groningen, The Netherlands; 2Departamento de Farmacologia, Escola Paulista de Medicina (EPM), Universidade Federal de São Paulo (UNIFESP), Rua 3 de Maio 100, Ed. INFAR, 3rd Floor, São Paulo 04044-020, Brazil

**Keywords:** crotamine, siRNA, syndecan-1, complement system, proximal tubular epithelial cells (PTECs)

## Abstract

Proteinuria drives progressive tubulointerstitial fibrosis in native and transplanted kidneys, mainly through the activation of proximal tubular epithelial cells (PTECs). During proteinuria, PTEC syndecan-1 functions as a docking platform for properdin-mediated alternative complement activation. Non-viral gene delivery vectors to target PTEC syndecan-1 could be useful to slow down alternative complement activation. In this work, we characterize a PTEC-specific non-viral delivery vector composed of the cell-penetrating peptide crotamine complexed with a syndecan-1 targeting siRNA. Cell biological characterization was performed in the human PTEC HK2 cell line, using confocal microscopy, qRT-PCR, and flow cytometry. PTEC targeting in vivo was carried out in healthy mice. Crotamine/siRNA nanocomplexes are positively charged, about 100 nm in size, resistant to nuclease degradation, and showed in vitro and in vivo specificity and internalization into PTECs. The efficient suppression of syndecan-1 expression in PTECs mediated by these nanocomplexes significantly reduced properdin binding (*p* < 0.001), as well as the subsequent complement activation by the alternative complement pathway (*p* < 0.001), as observed in either normal or activated tubular conditions. To conclude, crotamine/siRNA-mediated downregulation of PTEC syndecan-1 reduced the activation of the alternative complement pathway. Therefore, we suggest that the present strategy opens new venues for targeted proximal tubular gene therapy in renal diseases.

## 1. Introduction

Many renal diseases are associated with proteinuria. As proteinuria is independently associated with renal function impairment, anti-proteinuric treatment (mainly renin–angiotensin–aldosterone system intervention, eventually in combination with reduced salt intake) comprises a major cornerstone in clinical nephrology. Nevertheless, the elimination of proteinuria is very difficult, and most patients slowly progress towards renal failure. This indicates the need for additional treatment modalities, not only trying to reduce proteinuria even further, but also to reduce the harmful effects downstream of proteinuria. Importantly, proteinuria triggers a chronic activation of tubular cells, mainly of proximal tubular epithelial cells (PTECs) [1,2,3,4,5].

One of the molecules that shows a different expression profile in activated PTECs is syndecan-1, which is a major epithelial heparan sulfate proteoglycan [6,7,8]. Syndecan-1 acts as a co-receptor for growth factors and chemokines [9], as well as an autonomous endocytosis receptor [10,11]. Under physiological conditions, syndecan-1 is expressed at a low level and mainly in a basolateral pattern in PTECs [7]. During renal injury, syndecan-1 expression is increased and translocates from the basolateral to the apical membrane of PTECs, serving as a docking platform for heparin-binding proteins, including complement factors like properdin, thereby initiating the alternative pathway of the complement cascade [8]. We previously showed tubular and/or urinary properdin to be instrumental in progressive renal failure [8,12]. Therefore, targeting proximal tubular cells to reduce the expression of syndecan-1 can be a good approach to reduce the activation of the complement system, thereby decreasing or preventing renal injury progression [8,13,14]. This could be carried out using siRNA technology.

siRNA has great potential to treat a wide range of illnesses. Although naked siRNA has been applied in vivo, nanoparticle solutions are needed to increase tissue specificity and cell penetration properties, as well as to reduce endonuclease degradation and immunostimulating effects [15]. In the context of experimental acute kidney injury, investigations have predominantly targeted siRNA to PTECs up to now. PTEC-specific siRNA-containing nanoparticles were designed by electrostatic complex formation using modified chitosan polymers [16], polymeric CXCR4 antagonists [17], and functionalized carbon nanotubes [18], yielding spherical or elongated siRNA-containing particles. Alternatively, siRNA was encapsulated into poly(lactic-co-glycolic acid) mesoscale nanoparticles [19]. PTECs specificity was demonstrated to be via megalin or CXCR4-mediated endocytosis.

Crotamine is a positively charged cell-penetrating peptide (CPP) with 42 amino acid residues, which was originally isolated from the South American rattlesnake *Crotalus durissus terrificus* venom [20]. The ability of crotamine to be internalized by endocytosis with subsequent release from the endosomal/lysosomal vesicles confers a unique advantage for this cationic polypeptide as a non-viral gene carrier compared with any other known CPPs, whose endosomal/lysosomal entrapment is considered as a limiting factor for their use as a transfecting agent [21,22]. In addition, crotamine is non-toxic to cells in the micromolar range and, therefore, it can be safely used both in vitro and in vivo to transfect mammalian cells [20,21,22,23]. The proposed mechanism for internalization of crotamine or crotamine–nucleic acids complexes into cells involves its binding to cell surface heparan sulfate proteoglycans (HSPGs), which is needed for the uptake of this polypeptide by endocytosis [23,24,25,26,27]. After intraperitoneal (*ip*) administration in healthy mice, preferential accumulation of crotamine was observed in the kidney, more specifically in the brush border zone of proximal tubular epithelial cells (PTECs), with an internalization suggested to be mediated by syndecan-1 [27].

Combined, these unique characteristics of endo/lysosomal escape, together with the specificity for PTECs and renal safety in long-term administration, make crotamine an ideal nanocarrier of therapeutic molecules to actively target tubular cells. In this work, we evaluated and confirmed the viability of crotamine use as a nanocarrier of therapeutical syndecan-1 specific for siRNA into PTECs, not only by targeting siRNA into tubular cells based on the crotamine/syndecan-1 interaction, but also by reducing the properdin binding to this proteoglycan and, consequently, slowing down the alternative complement pathway activation.

## 2. Materials and Methods

### 2.1. Preparation and Characterization of Crotamine–siRNA Nanocomplexes

Crotamine was purchased from Smartox Biotechnology (Saint-Egrève, France—CRO001). The siRNA-targeting syndecan-1 mRNA GCCGCAATTGTGGCTACTAA, Allstars Negative Control siRNA, and fluorescent-labeled scrambled siRNA AF488 were from Qiagen (Hilden, The Netherlands—1027418, SI03650318, 1027292). All other reagents, when not specified in the text, were of analytical grade and were mainly purchased from Sigma-Aldrich Inc. (St. Louis, MO, USA).

### 2.2. Preparation and Characterization of Crotamine–siRNA Nanocomplexes

Samples with different crotamine–siRNA molar ratios were mixed in RNAse-free water by vortexing for 1 min at 37 °C, and the nanocomplexes formed by electrostatical force were then used immediately [28]. Alternatively, when needed, the complex growth was stopped by the addition of phosphate-buffered saline (PBS) or cell medium because, in both conditions, the presence of salt stopped further conjugation by limiting the electrostatical interactions between crotamine and siRNA, as previously demonstrated [29]. The nanocomplexes formed as described were characterized by electrophoresis mobility shift assay, dynamic light scattering (DLS), and transmission electron microscopy (TEM), and the stability of these nanocomplexes in the presence of serum was also assayed (for details on crotamine, siRNAs, and biophysical techniques, see Appendix A).

### 2.3. Crotamine–siRNA Complex Binding and Uptake in Cultured Cells

For crotamine–siRNA binding and uptake experiments, HK-2 cells (10^5^ cells/well) were plated on glass coverslips placed in a six-well plate, containing culture medium. The colocalization of the crotamine/siRNA complexes with syndecan-1 was investigated by a triple staining procedure, starting with the syndecan-1 labeling. The culture medium was removed and the cells were then incubated on ice with the primary antibody AF647 Alexa Fluor^®^ 647 mouse anti-human syndecan-1 (CD138; Bio-Rad/AbD Serotec, Hercules, CA, USA—MCA2459A647) (diluted 1:100 in culture medium) for 30 min, followed by washing with medium and incubation with the secondary antibody donkey anti-mouse IgG conjugated with AF647 (Invitrogen, Waltham, MA, USA—A32787) (1:100 in culture medium), also for 30 min, on ice. After washing with the culture medium, the cells were then incubated on ice with the complexes formed by fluorescently labeled AF555-crotamine and AF488-siRNA (at a molar ratio of 100:1, and with 50 nM of siRNA) for 30 min. After incubation, cells were washed with ice-cold PBS and fixed with 4% paraformaldehyde in PBS for 15 min at room temperature, followed by nuclear staining with 300 nM DAPI (4′,6-diamidino-2-phenylindole) for 10 min in the dark. Citifluor^TM^ AF1 Mounting Medium (EMS Acquisition Corp., San Francisco, CA, USA—17970) was used for embedment, before the confocal fluorescence microscopy analysis.

To evaluate the crotamine/siRNA internalization, HK-2 cells (10^5^ cells/well) were plated on glass cover slips placed in six-well plates. Then, the cells were maintained in cell culture medium and were incubated with the complexes formed by fluorescently labeled AF555-crotamine and AF488-siRNA (at a molar ratio of 100:1, and with 50 nM of siRNA) at 37 °C for 24 h. Then, the cells were incubated with 75 nM LysoTracker Deep Red (ThermoFisher Scientific Inc., Waltham, MA, USA—L12492) for lysosomes staining, while the nuclei were stained with 1 µg/mL of Hoechst 33258 (ThermoFisher Scientific Inc.—H3569) at 37 °C for 1 h. After this incubation, the cell medium was refreshed, cells were placed on ice, and live cells were analyzed by confocal fluorescence microscopy, following the method and conditions described in the Appendix A.

### 2.4. Downmodulation of Syndecan-1 Expression by the Crotamine–siRNA Complex in HK-2 Cells

The downregulation of syndecan-1 by the crotamine–siRNA nanocomplexes with crotamine–siRNA ratios of 50:1 and 100:1 was evaluated by adding these nanocomplexes with 3 nM of siRNA just before plating the HK-2 cells (10^5^ cells/well). Details on the siRNAs sequence used for syndecan-1 targeting can be found in the Appendix A. A scrambled siRNA Allstars (Qiagen) was used as a negative control and the transfection reagent lipofectamine 2000 (ThermoFisher Inc., #11668019) was used as a positive control. After the incubation for 48 h, cells were then incubated with 600 µL/well of non-enzymatic cell dissociation solution 1× (C5789—Sigma^®^) at 37 °C until cells were detached, then the cells were analyzed by flow cytometry to determine the expression of cell surface syndecan-1, properdin binding, and activated C3 deposition. The syndecan-1 mRNA levels were also quantified by real-time quantitative PCR (qRT-PCR) (as described in the Appendix A). In additional experiments, before the transfection mediated by crotamine/siRNA nanocomplexes, cells were also treated with doxorubicin for 24 h (as described in the Appendix A). All experiments were performed three times independently.

### 2.5. Crotamine/siRNA Complex Administration in Mice

Mice obtained from the Laboratory of Experimental Animals of the Center of Pharmacology and Molecular Biology (INFAR) of the Federal University of São Paulo (UNIFESP/EPM, SP, Brazil) were housed under controlled temperature and light regimen and with free access to food and water. The experiments with animals were performed following FELASA guidelines and this study was approved by the Ethic Committee for Animal Use of UNIFESP/EPM (No. 6237220116).

Male Swiss mice (10 weeks old, 25–30 g) were randomly divided into five experimental groups (N = 5 animals each): (1) control group receiving saline; (2) group for naked siRNA (0.1 mg/kg) injected 2 h before animal euthanasia; (3) group for naked siRNA (0.1 mg/kg) injected 24 h before animal euthanasia; (4) group for crotamine/siRNA complexes (at a molar ratio of 100:1, and with 50 nM of siRNA), administered in a dose of 0.1 mg/kg of siRNA, 2 h before animal euthanasia; and (5) group for crotamine/siRNA complexes (at a molar ratio of 100:1, and with 50 nM of siRNA), administered in a dose of 0.1 mg/kg of siRNA, 24 h before animal euthanasia. All mice received a single intraperitoneal (ip) injection in a final volume of 100 μL of saline as vehicle per animal. After euthanasia, the kidney, liver, and heart of the animals were collected and processed for subsequent histological analysis, as described in the Appendix A.

### 2.6. Statistical Analysis

In vitro data were expressed as values of the mean ± standard error of the mean (S.E.M), representing at least three independent experiments, and significance levels were analyzed using a parametric test (ANOVA with post-hoc Bonferroni test). The significance threshold adopted here was *p* ≤ 0.05. Data analyses were performed using the GraphPad Prism version 7.0 for Windows (GraphPad Software).

## 3. Results

### 3.1. Characterization of Crotamine/siRNA Nanocomplexes

Crotamine and siRNA complex formation was confirmed by the shift in migration of crotamine and siRNA in agarose gel electrophoresis, in which the negatively charged free siRNA migrates towards the anode (lower part of the gels) (Figure 1A,D) and the positively charged free crotamine migrates towards the cathode (upper part of the gels) (Figure 1B,E). However, the increasing crotamine/siRNA ratio (mol/mol) shifted the migration of formed complexes towards the cathode, leading to the increase in the proportion of siRNA complexed with crotamine, with an estimated rate of 75%, 90%, and 99% of siRNA complexed at ratios of 50:1, 100:1, and 200:1, respectively (Figure 1G). It is also important to notice that, at molar ratios of 100:1 and 200:1, the net charge of the formed complex was positive even in the presence of siRNA (Figure 1C,F).

Nanocomplexes’ size was monitored by dynamic light scattering (DLS), showing a smaller nanocomplex size with increased crotamine/siRNA ratios. On the other hand, the nanocomplex size increased over time (Figure 2A), while the size distribution of the nanocomplexes was homogeneous for all molar ratios evaluated here (Figure 2B). Transmission electron microscopy (TEM) allowed to characterize the structural details of the nanocomplexes, showing mono-disperse and uniformly spherical structures in general (Figure 2C). Therefore, TEM analysis corroborated the DLS measurements, showing smaller structures for increased molar ratios of crotamine/siRNA (Figure 2D). At 100:1 ratio, the mean size of the crotamine/siRNA particles was about 125 nm in the TEM analysis (Figure 2C,D).

The stability of the complex after incubation with human serum was evaluated by agarose gel electrophoresis, showing that the naked siRNA was rapidly degraded when exposed to serum nucleases, while the complex structure formed with crotamine protected siRNA from degradation (Figure 3).

### 3.2. Crotamine/siRNA Nanocomplexes’ Binding and Internalization into the PTEC HK2 Cell Line In Vitro and In Vivo

The interaction of unconjugated crotamine or crotamine/DNA complexes with HSPGs was demonstrated to be essential for their internalization by endocytosis [24]. As low temperatures were also demonstrated to prevent the rapid endocytosis of nanocomplexes [24], the interaction of the crotamine/siRNA nanocomplexes on the surface of HK-2 cells was also evaluated here at 4 °C. The fluorescently labeled crotamine and siRNA was colocalized in the surface of cultured HK-2 cells, confirming the integrity of the formed nanocomplex and confirming the interaction of this nanocomplex with the cell membrane (Figure 4A,B,F,J). The presence of syndecan-1 on the cell surface of HK-2 cells was confirmed by the immunostaining using the anti-syndecan-1 antibodies (Figure 4C), and its possible involvement in the nanocomplex anchoring was suggested by its colocalization with crotamine and siRNA fluorescent signals (Figure 4G,H,K,L). Most siRNA-positive pixels were triple positive for the fluorescence signals from the anti-syndecan-1 antibody and fluorescently labeled crotamine and siRNA (Figure 4I).

The translocation of siRNA into HK-2 cells mediated by crotamine was verified by incubating the nanocomplexes formed by the fluorescently labeled crotamine (AF555-crotamine, in red) and fluorescently labeled siRNA (AF488-siRNA, in green) with cultured HK-2 cells at 37 °C. Most of the siRNA fluorescence signal was observed to colocalize with crotamine at lysotracker-positive vesicles (in magenta) in the HK-2 cells’ cytoplasm (Figure 5), as a result of cellular uptake of the crotamine/siRNA complexes, in line with the suggestion of cellular uptake by endocytosis and accumulation in lysosomes, as previously demonstrated in the epithelial cell line derived from the ovary of the Chinese hamster (CHO cells) [24]. In addition, double-positive crotamine/siRNA complexes were also found to not colocalize with the lysotracker fluorescence signals. In the same way, siRNA fluorescent signal (in green), not colocalizing with crotamine (in red) or lysotracker (in magenta) fluorescent signals, was also found to be distributed in the cell cytosol (for relative quantification, see Figure 5M).

Analysis of the sections of kidney collected from mice 2 h after the administration of crotamine/siRNA nanocomplexes showed the siRNA fluorescent signal localized in the proximal tubules (Figure 6), in both the brush border and basolateral area of PTECs (Figure 6J–O). However, after 24 h of administration of the 100:1 ratio nanocomplex in healthy animals, siRNA was mainly internalized and accumulated in the cytosol of these cells (Figure 6P–R), confirming the successful targeting and in vivo uptake of the complex by PTECs. Evaluation of liver and heart allowed observing no significant fluorescence signals’ intensity relative to that observed in the kidney, indicating the specific accumulation in the kidneys, which is in good agreement with a previous biodistribution study conducted with radiolabeled native crotamine [30]. Kidneys of control mice injected with free fluorescently labeled siRNA (AF488-siRNA) showed no fluorescent signals (Figure 6A–I).

### 3.3. Downregulation of Syndecan-1 by Crotamine/siRNA Complexes Reduces Properdin Binding and Complement C3 Deposition in HK-2 Cells under Healthy and Diseased Conditions

Real-time PCR (qRT-PCR) analysis showed that crotamine/siRNA complexes reduced the expression of syndecan-1 mRNA mediated by the 100:1 ratio crotamine/siRNA complexes in HK-2 cells to about 50% relative to the mock-transfected cells, which were prepared using a scrambled siRNA (*p* < 0.01) (Figure 7A). Importantly, transfections mediated by the complexes formed by crotamine with scrambled siRNA (at same molar ratio, i.e., 100:1 ratio) showed no significant reduction in syndecan-1 mRNA transcripts, indicating the specificity of the syndecan-1 siRNA targeting without direct effects of crotamine on syndecan-1 expression. At the same molar concentration (3 nM) of syndecan-1 targeting siRNA, a reduction in syndecan-1 mRNA expression to about 25% compared with the control was observed for the transfections of syndecan-1 targeting siRNA with lipofectamine.

Moreover, analysis by flow cytometry also confirmed that crotamine/siRNA complexes were able to reduce the syndecan-1 expression in HK-2 cells to about 50% at both molar ratios evaluated here, namely at ratios of 50:1 (*p* < 0.001) and 100:1 (*p* < 0.01) (Figure 7B). The binding of the alternative complement factor properdin with HK2 cells is dependent on syndecan-1 [8], and a reduction to about 65% relative to mock-transfected cells was shown for the cells transfected with crotamine/siRNA nanocomplexes at a ratio of 50:1 (*p* < 0.001) and to about 35% for a molar ratio of 100:1 (*p* < 0.01) (Figure 7C). Following the properdin binding, the complement C3 deposition is also reduced in samples transfected by the crotamine/siRNA nanocomplexes upon incubation with serum, indicating a complement C3 deposition of about 42% relative to the control for cells transfected by crotamine/siRNA nanocomplexes, at molar ratios of 50:1 and 100:1 (*p* < 0.001 for both conditions) (Figure 7D).

The ability of crotamine/siRNA complexes to downmodulate the syndecan-1 expression in a condition in which this proteoglycan is overexpressed was also tested in a cellular model of nephrotoxicity. In Appendix A, we show the significant increase in syndecan-1 expression induced by doxorubicin, as determined by the mRNA and protein level measurements. Interestingly, crotamine/siRNA nanocomplexes reversed the overexpression of syndecan-1 induced by the treatment with doxorubicin, with a corresponding reduction in properdin binding and C3 activation back to the control levels.

## 4. Discussion

In this communication, we characterized non-viral crotamine/siRNA nanocomplexes and demonstrated the in vitro specificity and functional downmodulation of syndecan-1 and properdin-mediated alternative complement pathway activation by these complexes in PTECs. Moreover, we present here the targeting of crotamine/siRNA nanocomplexes towards PTECs in vivo following intraperitoneal (ip) injections in healthy mice.

The successful development of nanocomplexes formed by the conjugation of the cationic CPP crotamine with nucleic acid molecules was previously demonstrated by us [20]. These nanocomplexes spontaneously self-assemble in aqueous solutions, without the need for chemical modifications. At molar ratios of 100:1 and 200:1, even in the presence of siRNA, the net charge of the formed complex was positive, which is a characteristic described to be essential for the interaction of this complex with the cell membrane [28]. siRNAs are structures that are especially unstable and labile, making them highly susceptible to quick degradation in serum conditions. In the present work, we demonstrated that the siRNA remained intact after the formation of nanocomplexes with crotamine (crotamine/siRNA) even after incubation for 2 h with human serum, which present a high content of nucleases [31], indicating that the complex structure with crotamine is able to prevent the enzymatic degradation of siRNA by the blood serum nucleases. Packaging nucleic acids may serve as suitable RNase-resistant formulations for mRNA administration, and the siRNA protection is an important and necessary feature for a nanocomplex designed for systemic administration [31]. The nanocomplex formed by crotamine surrounding the siRNA, as previously suggested [29], may hamper the binding and recognition of this nucleic acid molecule by the nucleases present in the human serum, preventing its degradation, as shown here by the agarose gel analysis. However, we also have to consider that the presence of several proteins in the blood samples may have potentially contributed to the slight decrease in the siRNA band intensity in this gel, leading to a lower intensity smear that could not be detectable under our experimental conditions, as also observed in Figure 1 with a higher proportion of crotamine (i.e., ratios of 100:1 and 200:1).

In addition, the size of diameter of about 125 nm observed for crotamine/siRNA complexes, according to the literature, is the most acceptable size range for the use of nanocomplexes, depending on the employed delivery strategy in renal therapy [32,33,34,35]. The average size of the crotamine/siRNA complexes observed by TEM were suggested to be slightly different than that observed by DLS, in which the complex growing was observed over time and with no limitation on the dynamics of the natural growth of these complexes. Therefore, the addition of salt to stop the complex growing in the TEM experiments may have imposed some influence on the final estimated average sizes of the complexes. The temperature is also crucial for DLS measurements as the motion of macromolecules depends on their size, temperature, and solvent viscosity [36]. Additionally, there are also differences in the size distribution calculation of these techniques [37] and, in the present work, the aim was not to compare the analytical efficiency of these diverse techniques. However, for the ratio of 100:1, the initial size measured during DLS (143.7 ± 2.7 nm) is overall equivalent to the average size observed in TEM (125.0 ± 13.4 nm). Therefore, the ratio of 100:1 was mainly chosen for the further experiments of this work, because, at this ratio, the estimated complexation of about 90% of each molecule is observed, while maintaining a net positive charge. Furthermore, as demonstrated here, the nanocomplex at a ratio of 100:1, as well as at 50:1, may provide an acceptable size of the complex for an efficient cell internalization, while preventing risks of possible toxic effects associated with the use of higher concentrations of crotamine. Indeed, using both 50:1 and 100:1 nanocomplex ratios for the FACS experiments showed comparable cell viability of around 85% (by propidium iodide cell dead exclusion), as also seen in control cell cultures.

The ability of crotamine alone to form nanocomplexes to escape from the endo/lysosomal vesicles was previously demonstrated, and the delivery of siRNA in the cell cytoplasm, which are crucial attributes required for the therapeutic use of siRNA in an aim to silence its targeted complementary mRNA, was confirmed here not only by the fluorescence signals observed in the cell cytoplasm, but also by the demonstrated decrease in syndecan-1 expression in these cells. We, however, cannot exclude that crotamine/siRNA complexes or siRNA were partially trapped in EEA1 or TfR positive early endosomal vesicles, where a higher pH could preclude lysotracker positivity. However, more importantly, in our view, evidence of siRNA release into the cytosol comes mainly from the functional knock down experiments, which show a significant decrease in syndecan-1 expression compared with the respective controls, as described in the present work.

We have previously shown the specific targeting of crotamine towards PTECs in vivo, including data on bioavailability and toxicity [27]. Herein, we confirm the similar specificity and biodistribution of the crotamine/siRNA nanocomplexes that colocalized with syndecan-1 at the cell membrane of PTECs, as demonstrated here, confirming the role of this proteoglycan in crotamine interaction and internalization in this cell type [27].

Syndecan-1 is the main transmembrane HSPG present in epithelial cells and, during renal injury, syndecan-1 is overexpressed in PTECs [8,13], as previously demonstrated in proteinuric models in vivo [8]. The complex formed by crotamine and syndecan-1 targeting siRNA proved to effectively downmodulate the expression of syndecan-1 in PTECs in vitro, as measured at mRNA and protein levels. Importantly, the reduction in the expression of syndecan-1 in the cell membrane also reduces the binding of exogenous properdin, in a syndecan-1-dependent manner.

This finding is relevant in the context of proteinuric renal diseases, where plasma properdin now passes the leaky glomerular filter into the pro-urine and will bind with syndecan-1 on the apical membranes of PTECs. Considering that complement activation via the alternative pathway and C3 deposition on PTECs is dependent on properdin, being the single positive regulator of the alternative pathway [12], complement activation was also evaluated, showing a remarkably reduced activated C3 deposition in cells with reduced syndecan-1 expression. These present data corroborate the previous evidence showing that syndecan-1 acts as a docking platform for alternative pathway activation via properdin [8,38]. Proximal tubular alternative complement activation may also trigger inflammatory responses contributing to tubulointerstitial injury and the progression of renal damage [8,12,39].

The use of a crotamine/siRNA complex to downregulate syndecan-1 expression could be a good strategy for reducing properdin binding, preventing alternative complement activation and, consequently, mitigating proteinuric renal damage. Therefore, the efficacy of the crotamine/siRNA nanocomplex in gene silencing was evaluated not only under physiological and healthy conditions, but also after doxorubicin treatment, which leads to syndecan-1 overexpression, mimicking a pathological condition. Remarkably, the nanocomplex was thus able to downmodulate syndecan-1 expression even in aberrant conditions, and the same reduction pattern was also observed for properdin binding and C3 deposition in doxorubicin-treated cells. Therefore, the present data reinforce the strong correlation involving the presence of syndecan-1 and the properdin binding and alternative complement activation.

Ultimately, 2 h after the injection of the fluorescently labeled crotamine/siRNA complex in mice, through the ip route, the presence of siRNA was observed in proximal tubule, on both the apical and basolateral sides of PTECs. The apical detection might be due to smaller-sized crotamine/siRNA nanocomplexes or due to free siRNA eluted from the complexes, because of the anionic competition of negatively charged constituents in the glycocalyx and glomerular basement membrane of the glomerular capillary wall, which would be required to pass the glomerular barrier filtration. The crotamine/siRNA complexes detected in the basolateral membrane by its turn more likely correspond to the intact 125 nm complexes transcytosed across peritubular capillaries. After 24 h, siRNA was also detected in the cytoplasm of PTECs, with a distribution pattern similar to that already demonstrated for free crotamine in the kidney of mice, following administration by the ip route. Therefore, the present data are in line with several pieces of evidence showing that nanoparticles can be specifically targeted to PTECs in vivo [17,18,19]. Importantly, in a kidney disease state, the glomerular basement membrane meshes might become larger [40], thus facilitating the passage of molecules and/or conjugates larger than 100 nm, hypothetically enabling a more effective PTEC targeting.

The exceptional features of crotamine or complexes formed with crotamine, including the easy assembly, endo/lysosomal escape, and unique specificity for targeting PTECs, make their adoption in tubular injuries or tubulopathies a promising strategy. Therapies adopting siRNA have the advantage of providing a transient knockdown, which is indicated when a temporary change in gene expression is desired [41]. Crotamine could be potentially used to deliver cargos to the proximal tubules after conjugation with other molecules with putative therapeutic properties. For instance, antioxidants such as superoxide dismutase (SOD) are known to inhibit the progression of acute renal failure [42]. Rho kinase inhibitors could be used to prevent diabetic nephropathy progression [43]. Lastly, multikinase inhibitors such as sunitinib were demonstrated to play a role in protection against tubulointerstitial fibrosis [44,45]. However, experimental demonstrations are still needed to confirm the translational value of these potential applications.

In conclusion, the crotamine/siRNA complex was demonstrated here to be able to downregulate the expression of the proteoglycan syndecan-1 in PTECs in vitro at both mRNA and protein levels in physiological and nephrotoxic conditions. This expression reduction consequently decreased the properdin binding to syndecan-1 and the subsequent deposition of the complement factor C3, thus preventing alternative complement activation. The crotamine/siRNA complex was able to target PTECs in healthy mice with the same specificity already described for unconjugated free crotamine. Furthermore, this specific targeting for PTECs opens new paths for a plethora of possible therapeutic interventions in tubulopathies with the potential to be translated into clinical application.

## Figures and Tables

**Figure 1 pharmaceutics-15-01576-f001:**
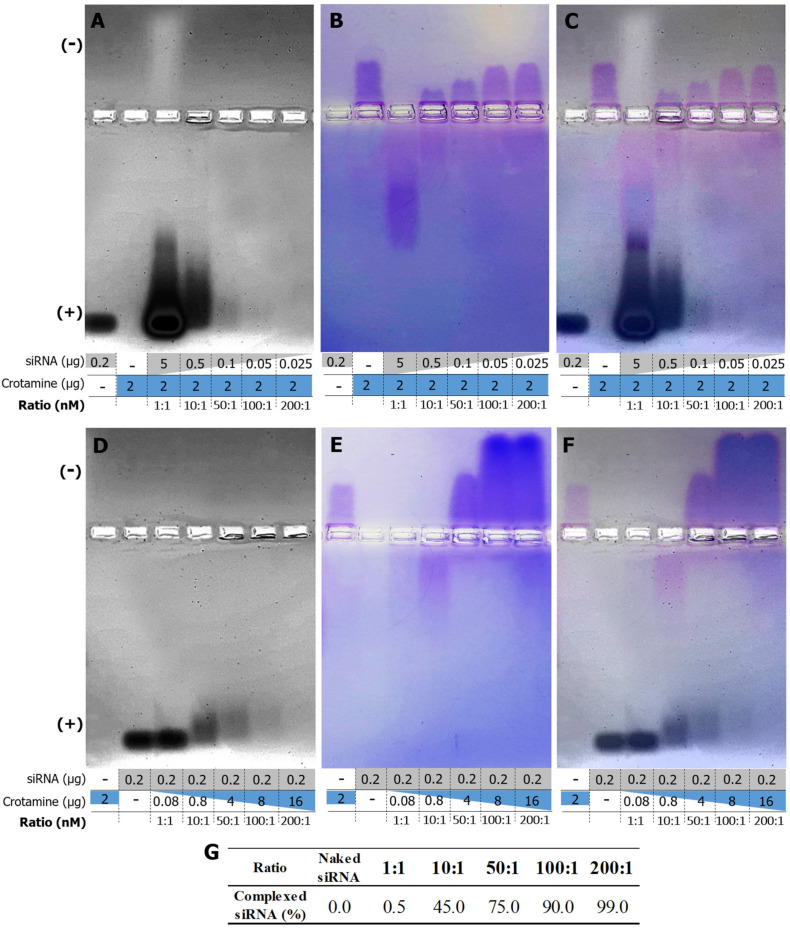
Electrophoretic mobility analysis of the crotamine/siRNA nanocomplexes in non-denaturing conditions. Various ratios were analyzed on a 1% non-denaturing agarose gel maintaining the concentration of the peptide crotamine while varying the concentration of siRNA (**A**–**C**) and varying the concentration of the peptide for a fixed amount of siRNA (**D**–**F**). The gels were stained with ethidium bromide to visualize siRNA (**A**,**D**) and with Coomassie brilliant blue R to visualize crotamine (**B**,**E**). Overlay images showing both crotamine and siRNA mobility (**C**,**F**). The intensity of the siRNA bands (**D**) was used to estimate the rate of siRNA complexed with crotamine (**G**). Images are representative of two independent assays.

**Figure 2 pharmaceutics-15-01576-f002:**
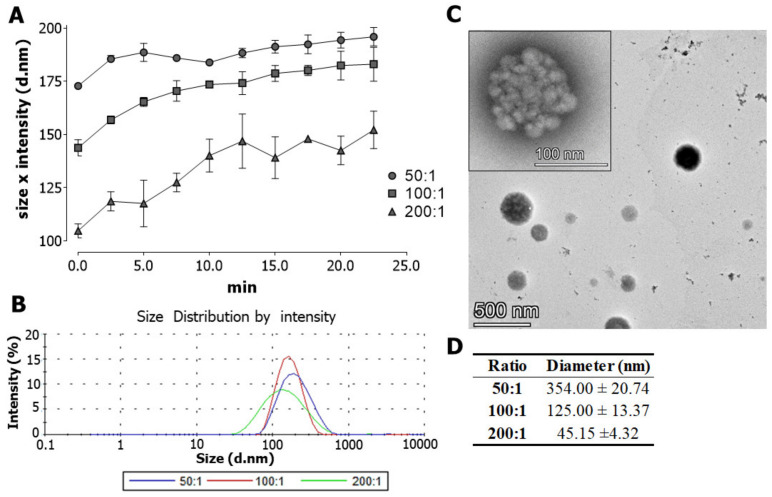
Size characterization of crotamine/siRNA nanocomplexes. Dynamic light scattering (DLS) measurement of the nanocomplexes over time at different molar ratios (50:1, 100:1, and 200:1) (**A**). Size distribution spectrum determined by Zetasizer Nano ZS (at ratios of 50:1, 100:1, and 200:1) (**B**). Transmission electron microscopy (TEM) images showing the size and shape of particles at a molar ratio of 100:1 (**C**). Table with the diameter of particles at different molar ratios (50:1, 100:1, and 200:1) analyzed by TEM. Data represented as mean ± SEM (**D**). Scale bars = 100 and 500 nm. Images are representative of three independent assays.

**Figure 3 pharmaceutics-15-01576-f003:**
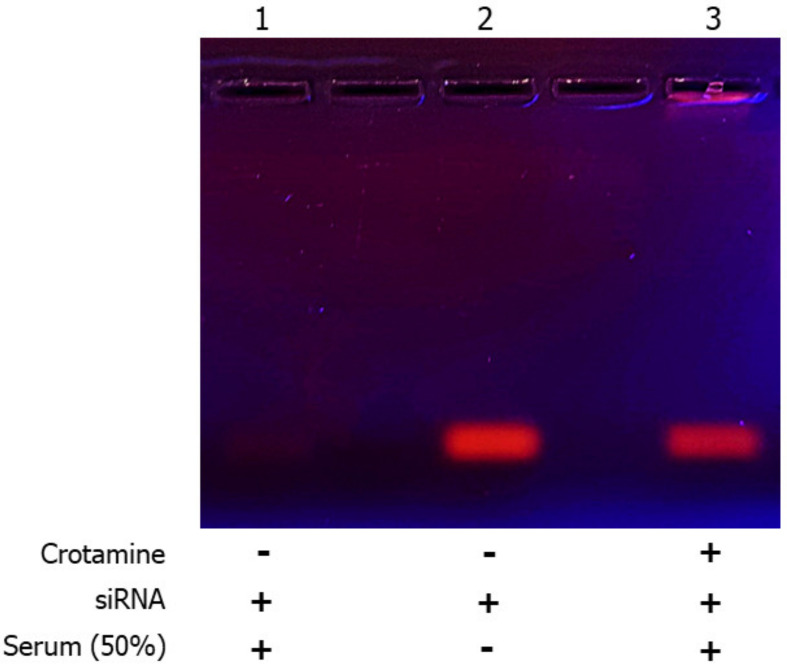
Stability of the crotamine/siRNA nanocomplexes to degradation in serum. Incubation of naked/unconjugated siRNA (1,2) or crotamine/siRNA nanocomplex (3) at a molar ratio of 50:1 with fetal bovine serum (FBS) (final concentration of 50% *v*/*v*) at 37 °C (1,3) for 2 h. SDS (0.5%) was added to displace the siRNA from the nanocomplex, and aliquots were then analyzed by electrophoresis in 1% agarose gel. The image is representative of two independent assays.

**Figure 4 pharmaceutics-15-01576-f004:**
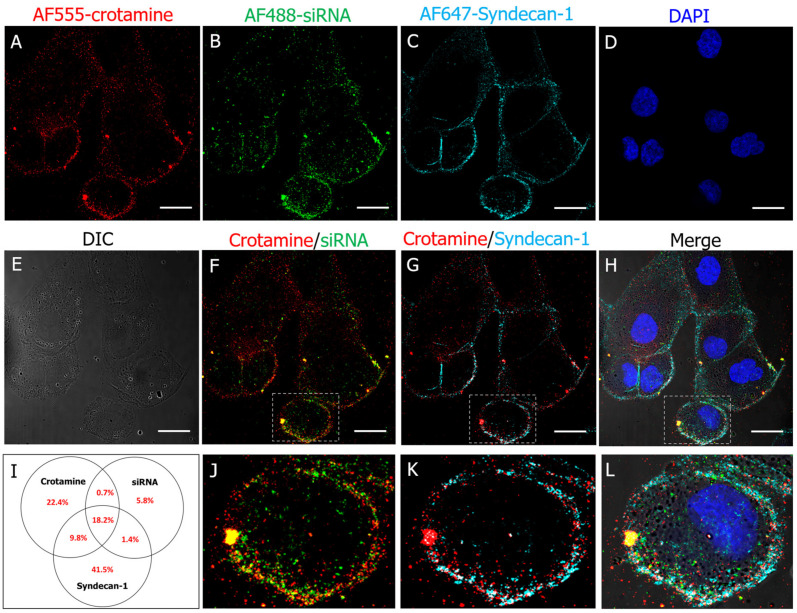
Colocalization of syndecan-1 and crotamine/siRNA nanocomplexes on the cell surface of PTECs in vitro by multicolor confocal immunofluorescence microscopy. Human PTEC HK-2 cells were incubated with fluorescently labeled AF555-crotamine/AF488-siRNA complexes formed at a molar ratio of 100:1 with 50 nM of siRNA, at 4 °C for 30 min (**A**,**B**). Immunorecognition of syndecan-1 by mouse anti-human syndecan-1 (Alexa647) (**C**). Nuclei stained with DAPI (blue) (**D**,**H**,**L**). Differential interference contrast (DIC) image (**E**) and overlay of AF555-crotamine (red) and AF488-siRNA (green) (**F**,**J**). Overlay of AF555-crotamine (red) and AF647-syndecan-1 (cyan) (**G**,**K**), and image merge with all four channels including DIC (**H**,**L**). Magnified images of overlays are also shown (**J**–**L**). Colocalization was analyzed using the JacoP plugin for ImageJ software (1.44 version) based on Mander’s coefficient. Venn diagram values represent percentage positive pixels for (overlapping) stainings, measured in 20 cells per calculation (**I**). Bars = 25 μm. Images are representative of three independent assays. Objective 63× oil.

**Figure 5 pharmaceutics-15-01576-f005:**
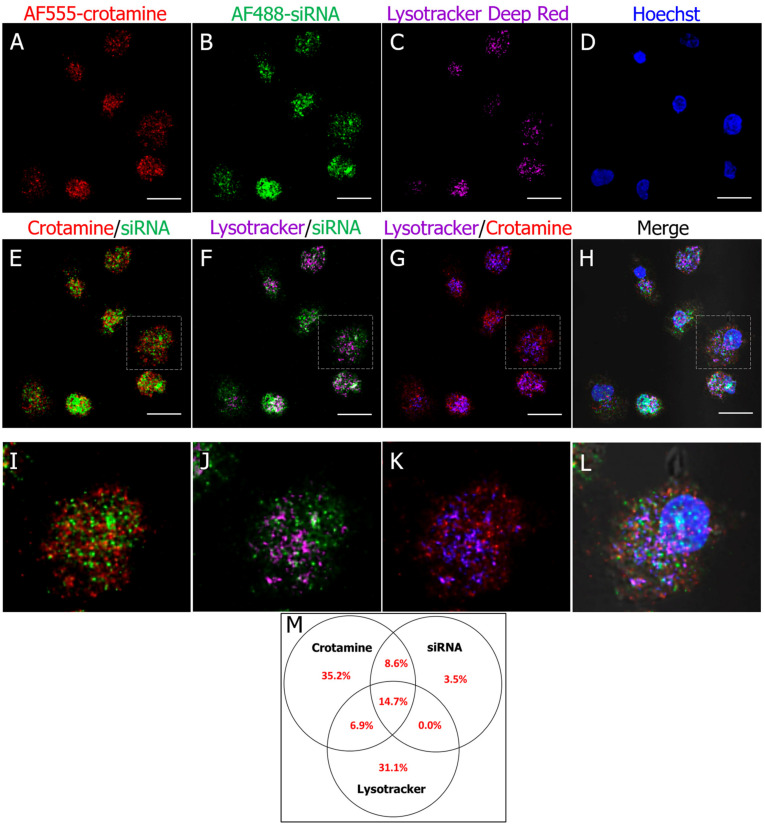
Multicolor confocal immunofluorescence microscopy showing internalization of crotamine/siRNA nanocomplexes by PTECs in vitro and colocalization with lysosomes in cell cytoplasm. HK-2 cells incubated with fluorescently labeled AF555-crotamine/AF488-siRNA complexes at a molar ratio of 100:1 with 50 nM of siRNA, at 37 °C for 24 h. AF555-crotamine (red) (**A**,**E**,**G**,**H**,**I**,**K**,**L**), AF488-siRNA (green) (**B**,**E**,**F**,**H**,**I**,**J**,**L**), lysosomes (magenta) (**C**,**F**,**G**,**H**,**J**,**K**,**L**), and nuclei stained with Hoechst (blue) (**D**,**H**,**L**). Overlay of crotamine and siRNA (**E**,**I**), overlay of lysosomes and siRNA (**F**,**J**), overlay of lysosomes and crotamine (**G**,**K**), and merged channels with differential interference contrast (DIC) images (**H**,**L**). Magnified overlay images (**I**–**L**). Colocalization was analyzed using the JacoP plugin for ImageJ software (1.44 version) based on Mander´s coefficient. Venn diagram values represent percentage positive pixels for (overlapping) stainings, measured in 20 cells per calculation (**M**). Bar = 20 μm. Images are representative of three independent assays. Objective 63× oil.

**Figure 6 pharmaceutics-15-01576-f006:**
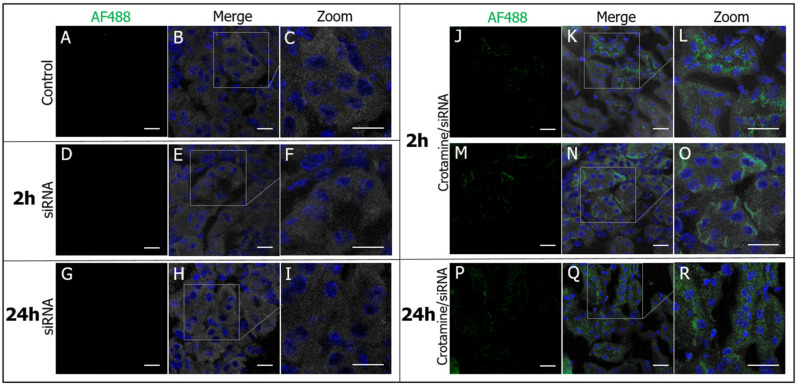
Localization of crotamine/siRNA nanocomplexes in PTECs of mice kidneys. Kidney sections from mice receiving the vehicle saline (**A**–**C**), naked AF488-siRNA (**D**–**I**), or a single *ip* administration of the complex formed by crotamine and fluorescently labeled AF488-siRNA at a molar ratio of 100:1 and siRNA concentration of 0.1 mg/kg, injected 2 h (**J**–**O**) or 24 h (**P**–**R**) before the animal euthanasia. Confocal immunofluorescence microscopy of AF488-siRNA stained in green (**A**,**D**,**G**,**J**,**M**,**P**), overlay of green and nuclei stained with DAPI (blue), and differential interference contrast (DIC) showing the kidney tissue structure (**B**,**E**,**H**,**K**,**N**,**Q**) and magnified overlay images (**C**,**F**,**I**,**L**,**O**,**R**). Bar = 20 μm. Images are representative of three independent assays. Objective 100× oil.

**Figure 7 pharmaceutics-15-01576-f007:**
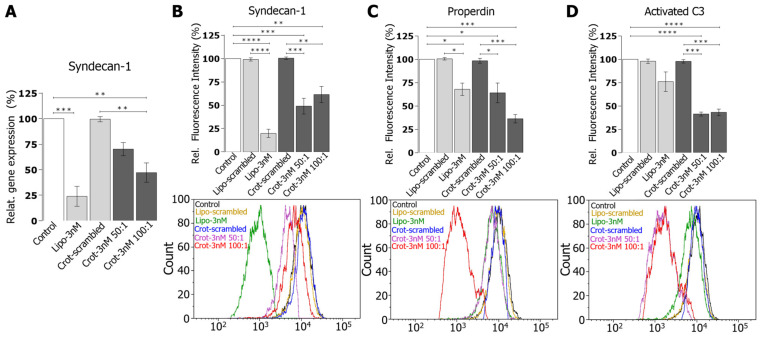
Downmodulation in vitro of syndecan-1 expression by crotamine/siRNA nanocomplexes, with decreased properdin binding and C3 complement activation in PTECs. qPCR quantification of syndecan-1 expression normalized against GAPDH (**A**); flow cytometry analysis of syndecan-1 protein expression, with a respective representative flow cytometry chart shown below (**B**); flow cytometry analysis of the properdin binding with PTECs, with a respective representative flow cytometry chart shown below (**C**); and flow cytometry analysis of activated C3 complement factor, with a respective representative flow cytometry chart shown below (**D**). Data represented as mean ± SEM % of relative gene expression determined by qPCR and mean ± SEM % relative fluorescence intensity monitored by flow cytometry (data were normalized for the untreated control, which was considered as 100%). Data were statistically analyzed by ANOVA with post-hoc Bonferroni test (* *p* ≤ 0.05, ** *p* ≤ 0.01, *** *p* ≤ 0.001, **** *p* ≤ 0.0001) (N = 3).

## Data Availability

The data underlying this article are available in the article and in its online Appendix A. Raw data are available upon request.

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
