# Peer review of "Crotamine/siRNA Nanocomplexes for Functional Downregulation of Syndecan-1 in Renal Proximal Tubular Epithelial Cells"

_pharmaceutics, 2023, doi:10.3390/pharmaceutics15061576_

Round 1
Reviewer 1 Report
In this research article, the authors presented “Crotamine/siRNA Nanocomplexes for Functional Downregulation of Syndecan-1 in Renal Proximal Tubular Epithelial Cells”. From my point of view, the topic is fascinating. The manuscript is concise and well-written. However, it has many issues to be taken care of before its publication in Pharmaceutics.
Following are my suggestions:
1) Authors should add the product number of the chemicals they have used. It will help others to repeat the synthesis.
2) Authors can move figure 1 a-c to the supporting info. It is not required in the main manuscript and creating unnecessary confusion.
3) Why the authors chose the Crotamine/siRNA up to 100:1 in further experiments?
4) The 200:1 particles are small and should be better for delivery. Why have the authors not used them for further studies?
5) It is not enough to provide fluorescence images for in vivo delivery applications. The authors could have performed other experiments to support their claims.
Author Response
First of all, we would like to thank the reviewer for time invested in our work. We appreciate the suggestions and try to answer all inquiries below.
Comments and Suggestions for Authors
In this research article, the authors presented “Crotamine/siRNA Nanocomplexes for Functional Downregulation of Syndecan-1 in Renal Proximal Tubular Epithelial Cells”. From my point of view, the topic is fascinating. The manuscript is concise and well-written. However, it has many issues to be taken care of before its publication in Pharmaceutics.
Following are my suggestions:
- Authors should add the product number of the chemicals they have used. It will help others to repeat the
The text was revised and corrected accordingly. Thank you!
2) Authors can move figure 1 a-c to the supporting info. It is not required in the main manuscript and creating unnecessary confusion.
We are very appreciative for the reviewer observation on this matter, but we need to remark that panel a-c and panel d-f bring different and important information. In panel a-c we can observe the effect of the excess of siRNA to the complex in different ratios. While in panel d-f we observe the effect of excess of peptide to the same ratios. Therefore we would like to keep the Figure as it is.
3) Why the authors chose the Crotamine/siRNA up to 100:1 in further experiments?
We are grateful for the reviewer remark on this matter, as this gave us the opportunity to better discuss this topic. The following information was added to the discussion topic: “The ratio of 100:1 was elected for the further experiments of this work, as delivers an acceptable size for cell internalization while preventing toxic effects of higher concentrations of crotamine. Furthermore, at this ratio, the complexation is about 90% while remaining a net positive charge.”
4) The 200:1 particles are small and should be better for delivery. Why have the authors not used them for further studies?
The ratio of 200:1 requires a higher concentration of crotamine which might increase the probability of toxic effects. This information was clarified in the discussion topic: “The ratio of 100:1 was elected for the further experiments of this work, as delivers an acceptable size for cell internalization while preventing toxic effects of higher concentrations of crotamine. Furthermore, at this ratio, the complexation is about 90% while remaining a net positive charge.”
5) It is not enough to provide fluorescence images for in vivo delivery applications. The authors could have performed other experiments to support their claims.
Thank you for this observation as this gave us the opportunity to revisit this topic. We added to the Results section the following: Other organs were checked and no significant signal was observed when compared to kidney signal intensity, indicating the high specific accumulation in this organ. Previous studies of our group with crotamine either complexed or non-complexed with DNA support this information (Crotamine mediates gene delivery into cells through the binding to heparan sulfate proteoglycans - PubMed (nih.gov), Long term safety of targeted internalization of cell penetrating peptide crotamine into renal proximal tubular epithelial cells in vivo - PubMed (nih.gov)).
Reviewer 2 Report
After careful reading of the manuscript entitled “Crotamine/siRNA Nanocomplexes for Functional Downregulation of Syndecan-1 in Renal Proximal Tubular Epithelial Cells” I have the following comments:
1. Words like “in vitro”, “in vivo” and “per” should be written in italic
2. Materials and methods section 2.2. authors reported “Samples with different crotamine-siRNA molar ratio (nM) were mixed in RNAse free water to produce the nanocomplexes” please indicate the incubation time and temperature.
3. The sequences of all used siRNA should be mentioned in materials section.
4. The methods written in section 2.4 are different than those written in Supplementary information, which are well described. Please revise.
5. Section 3.1. authors reported “Ratios of 50:1, 100:1 and 200:1 showed that increasing amounts of siRNA was complexed with crotamine with approximately 75, 90 and 99% of siRNA complexed” however, the amount of siRNA is fixed as indicated in the legend of Figure 1 “The intensity of the siRNA bands (D) was used to estimate the rate of siRNA complexed with crotamine….” Please revise.
6. There is a difference between nanocomplex size measured by the DLS and by TEM. Could you explain?
7. Authors should comment the DLS results in details.
8. Values in section 3.3 should be presented as mean ± SEM %
9. Results section in Supplementary data. “ H-2 cells” please correct.
10. Did you verify that your nanocomplex does not exist in other organs rather than kidney?
Author Response
First of all, we would like to thank the reviewer for time invested into our work. We appreciate the comments and suggestions and try to answer all inquiries below.
Comments and Suggestions for Authors
After careful reading of the manuscript entitled “Crotamine/siRNA Nanocomplexes for Functional Downregulation of Syndecan-1 in Renal Proximal Tubular Epithelial Cells” I have the following comments:
- Words like “in vitro”, “in vivo” and “per” should be written in italic
The text was revised and corrected accordingly. Thank you!
- Materials and methods section 2.2. authors reported “Samples with different crotamine-siRNA molar ratio (nM) were mixed in RNAse free water to produce the nanocomplexes” please indicate the incubation time and temperature.
We are grateful for the reviewer observation. The topic was clarified in Method section.
- The sequences of all used siRNA should be mentioned in materials section.
We are grateful for the reviewer observation. The topic was clarified in Method section.
- The methods written in section 2.4 are different than those written in Supplementary information, which are well described. Please revise.
In the main manuscript, section 2.4, we described the set-up of the HK-2 transfection experiments. Details on doxorubicin treatment, siRNA used, RT-PCR and flow cytometry are given in the Supplemental file. So, no differences, just addition of crucial details are given in the Supplementary file.
- Section 3.1. authors reported “Ratios of 50:1, 100:1 and 200:1 showed that increasing amounts of siRNA was complexed with crotamine with approximately 75, 90 and 99% of siRNA complexed” however, the amount of siRNA is fixed as indicated in the legend of Figure 1 “The intensity of the siRNA bands (D) was used to estimate the rate of siRNA complexed with crotamine….” Please revise.
We are very appreciative for the reviewer remark. The text was revised and corrected accordingly: “Rising the crotamine/siRNA ratio (mol/mol) increased the mobility of the complexes migrating towards the cathode, and also the amount of siRNA complexed with crotamine with approximately 75, 90 and 99% of siRNA complexed for ratios of 50:1, 100:1 and 200:1, respectively (Figure 1G).” Thank you!
- There is a difference between nanocomplex size measured by the DLS and by TEM. Could you explain?
We are grateful for the reviewer observation. The topic was clarified in Discussion section: “The average size of the crotamine/siRNA complexes observed by TEM suggested to be slightly different than that observed by DLS, although the addition of salt to stop the complex growing in the TEM experiments may impose some influence in this matter, as in DLS, the complex growing was observed over time and with no limitation in the dynamics of natural growth of these complexes. Additionally, there are also differences in the size distribution calculation of these techniques. However, for the ratio of 100:1, the initial size measured during DLS (143.7 ± 2.7 nm) is equivalent to the average size observed in TEM (125.0 ± 13.4).”
- Authors should comment the DLS results in details.
We are very appreciative for the reviewer observation on this matter, this topic was clarified in Discussion section as indicated in previous question raised by the reviewer.
- Values in section 3.3 should be presented as mean ± SEM %
We are grateful for the reviewer remark on this matter. Indeed, the data is presented as mean ± SEM % of relative gene expression determined by qPCR and mean ± SEM % of relative fluorescence intensity monitored by flow cytometry (data were normalized for the untreated control, which was considered as 100%). The normalization to control is required to provide a stable reference point against which the measurements can be referred.
The text was revised and corrected accordingly. Thank you!
- Results section in Supplementary data. “ H-2 cells” please correct.
The text was revised and corrected accordingly. Thank you!
- Did you verify that your nanocomplex does not exist in other organs rather than kidney?
We are grateful for the reviewer remark on this matter, as this gave us the opportunity to revisit this topic. We added to the Results section the following: Other organs were checked and no significant signal was observed when compared to kidney signal intensity, indicating the high specific accumulation in this organ. Previous studies of our group with crotamine either complexed or non-complexed with DNA support this information (Crotamine mediates gene delivery into cells through the binding to heparan sulfate proteoglycans - PubMed (nih.gov), Long term safety of targeted internalization of cell penetrating peptide crotamine into renal proximal tubular epithelial cells in vivo - PubMed (nih.gov)).
Reviewer 3 Report
The manuscript entitled “Crotamine/siRNA nanocomplexes for functional downregulation of syndecan-1 in renal proximal tubular epithelial cells” by Joana D`Arc Campeiro et al. designed and formulated a nanocomplex composed a cell penetration peptide-crotamine and a siRNA targeting to Syndecan-1 gene for proximal tubular epithelia cells (PTECs) delivery and gene regulation in complement pathway. It demonstrated that crotamine/siRNA complex can internalize into PTECs and downregulate Syndecan-1 gene expression efficiently by a series of in vitro functional studies. After IP injection in mouse model, those nanocomplexes were observed to accumulate in kidney tissues by histological staining study. A relative comprehensive investigation on the property and function of crotamine/siRNA nanocomplexes in this manuscript. However, following questions still need to be addressed:
1. In Introduction section, the authors should include the current research progress and the barricade for PTECs delivery to give an overview before introducing crotamine as a potential targeting ligand.
2. The detail of design and preparation of crotamine/siRNA is lacking. It’s not clear how the nanocomplexes form, by chemical conjugation or by electrostatic force? The authors mentioned that the complexes are made by vertex and then used immediately at line 77-78. It raises a question that how stable the nanocomplexes are in a regular storage condition. “by vortex” is too general, any speed limit or time requirement? And at line 76, it’s not appropriate to use “(nM)” as the unit for “molar ratio”.
3. Please include the siRNA sequence in the method part. How is the siRNA sequence selected? Is there any chemical modification on siRNA?
4. In Figure1, the authors attempted to characterize crotamine/siRNA complex by gel electrophoresis. From the gel images, there’s not a uniform and overlapping band of siRNA and crotamine. Are the complexes dissociated under the electrophoresis condition? Figure 1G provides the estimate siRNA complexed rate, but it is not accurate since all the gel bands of siRNA look very smear when complexed with crotamine.
5. In Figure2A, it shows that the size of crotamine/siRNA increased over the time. Is it happened before or after adding the salt based stop buffer? And please provide the rationale why the nanocomplex has a smaller size with higher crotamine/siRNA ratio. What’s the PDI of the nanocomplex? Figure 2C indicates that these nanocomplexes are in spherical shape, how can they from sphere? Are these particles homogeneous?
6. In Figure 3, the band intensity of lane3 is slightly lower than lane2. Is it caused by partial siRNA degradation? Have the authors studied the serum stability with longer incubation time? And what’s the mechanism that crotamine/siRNA complex can help improve the stability?
7. For cell uptake and colocalization study shown in Figure 4, why the incubation was done at 4 degree rather than 37 degree condition? The endocytosis process is energy driven so the low temperature will slow down the cell uptake in theory.
8. How many cells taken into the analysis to calculate %positive overlapping staining in Figure 4I and 5M?
9. Are the crotamine/siRNA nanocomplexes biocompatible? At least, a cell cytotoxicity study should be conducted in vitro to evaluate the safety.
10. Figure 6 well demonstrated that crotamine/siRNA nanocomplexes can accumulate into kidney tissues after 2hr IP injection. Have the authors look at the distribution in other tissues? And what’s the knockdown level of Syndecan-1 in these kidney tissues?
Author Response
First of all, we would like to thank the reviewer for time invested into our work. We appreciate the remarks and suggestions and try to answer all inquiries below.
Comments and Suggestions for Authors
The manuscript entitled “Crotamine/siRNA nanocomplexes for functional downregulation of syndecan-1 in renal proximal tubular epithelial cells” by Joana D`Arc Campeiro et al. designed and formulated a nanocomplex composed a cell penetration peptide-crotamine and a siRNA targeting to Syndecan-1 gene for proximal tubular epithelia cells (PTECs) delivery and gene regulation in complement pathway. It demonstrated that crotamine/siRNA complex can internalize into PTECs and downregulate Syndecan-1 gene expression efficiently by a series of in vitro functional studies. After IP injection in mouse model, those nanocomplexes were observed to accumulate in kidney tissues by histological staining study. A relative comprehensive investigation on the property and function of crotamine/siRNA nanocomplexes in this manuscript. However, following questions still need to be addressed:
- In Introduction section, the authors should include the current research progress and the barricade for PTECs delivery to give an overview before introducing crotamine as a potential targeting ligand.
We are grateful for the reviewer observation. The topic was clarified in an extra paragraph in the Introduction section. Barricades for PTECs delivery, especially the glomerular filtration barrier, are mentioned in the Discussion part of the manuscript.
- The detail of design and preparation of crotamine/siRNA is lacking. It’s not clear how the nanocomplexes form, by chemical conjugation or by electrostatic force? The authors mentioned that the complexes are made by vertex and then used immediately at line 77-78. It raises a question that how stable the nanocomplexes are in a regular storage condition. “by vortex” is too general, any speed limit or time requirement? And at line 76, it’s not appropriate to use “(nM)” as the unit for “molar ratio”.
We are very appreciative for the reviewer observation on this matter, the topic was clarified in Method section.
- Please include the siRNA sequence in the method part. How is the siRNA sequence selected? Is there any chemical modification on siRNA?
We are very appreciative for the reviewer observation on this matter, the topic was clarified in Method section.
- In Figure1, the authors attempted to characterize crotamine/siRNA complex by gel electrophoresis. From the gel images, there’s not a uniform and overlapping band of siRNA and crotamine. Are the complexes dissociated under the electrophoresis condition? Figure 1G provides the estimate siRNA complexed rate, but it is not accurate since all the gel bands of siRNA look very smear when complexed with crotamine.
We are grateful for the reviewer remark on this matter. According to Rio DC 2014 (doi:10.1101/pdb.prot080721): “The electrophoretic mobility shift assay (EMSA), or gel mobility shift assay, is a popular and powerful technique for the detection of RNA–protein interactions. It relies on the fact that naked RNA has certain mobility on nondenaturing gels, but if the RNA is bound by protein, the mobility of the RNA is reduced. Therefore, the binding of protein results in a characteristic upward shift of the RNA on a gel.” The complexes did not dissociate under experimental conditions, but stainability of siRNA is progressively reduced under increasing complex formation with crotamine. The crotamine/siRNA interaction is confirmed by the shift occurring in both bands. In Figure 1G the siRNA complexed rate was estimated based on the signal intensity of each siRNA band in Figure 1D when compared to the siRNA band in absence of crotamine (100% intensity, but 0% complexed).
- In Figure2A, it shows that the size of crotamine/siRNA increased over the time. Is it happened before or after adding the salt based stop buffer? And please provide the rationale why the nanocomplex has a smaller size with higher crotamine/siRNA ratio. What’s the PDI of the nanocomplex? Figure 2C indicates that these nanocomplexes are in spherical shape, how can they from sphere? Are these particles homogeneous?
We are grateful for the reviewer remark on this matter. The nanocomplex increasing overtime happens without the salt addition as the presence of salt stops the complex formation by altering the electrostatical interactions. This information was clarified in Methods section. Increasing the crotamine/siRNA ratio progressively precludes siRNA molecules to form bridges with crotamine complexed to other siRNA molecules, thereby reducing the formation of larger complexes. PDI of the complexes was always <0,2 as can be seen from the DLS distribution curve. Yes, the complexes were always spherical in form, which apparently is the most favorable form of the siRNA/crotamine complexes formed by electrostatic interactions. Homogeneity of the nanoparticles is shown by distribution of the DLS analysis in 2B and by the small SEM in 2D.
- In Figure 3, the band intensity of lane3 is slightly lower than lane2. Is it caused by partial siRNA degradation? Have the authors studied the serum stability with longer incubation time? And what’s the mechanism that crotamine/siRNA complex can help improve the stability?
We are grateful for the reviewer observation. In fact, the lower intensity in lane 3 might be caused by the partial siRNA degradation, as for this experiment the crotamine/siRNA adopted was 50:1 corresponding to about 75% of siRNA complexed. We did not evaluated the serum stability with longer incubation time as previous studies conducted by the group indicated that for in vitro studies crotamine is internalized by the cells in less than 10 minutes and for in vivo studies in mice, crotamine reaches the kidney in less than 2 hours (Crotamine mediates gene delivery into cells through the binding to heparan sulfate proteoglycans - PubMed (nih.gov), Long term safety of targeted internalization of cell penetrating peptide crotamine into renal proximal tubular epithelial cells in vivo - PubMed (nih.gov)).The siRNA remained intact after exposition of crotamine/siRNA complex to nucleases present in human serum, indicating that the complex structure with crotamine might be preventing the enzymatic degradation of siRNA by nucleases present in serum. We hypothesize that crotamine complexes with siRNA and the excess of crotamine forms a positively charged capsule around the structure protecting the siRNA in the inner part of the nanocomplex structure.
- For cell uptake and colocalization study shown in Figure 4, why the incubation was done at 4 degree rather than 37 degree condition? The endocytosis process is energy driven so the low temperature will slow down the cell uptake in theory.
We are grateful for the reviewer remark on this matter. In fact the incubation at 4 degree was deliberately used to prevent the endocytosis process being able to confirm that the nanocomplex is internalized by this mechanism as it was already shown for other crotamine-complexes (Nascimento et al., 2007; Hayashi et al., 2008; Hayashi et al., 2012). Additionally, the experiment allowed us to evaluate the interaction between the complex and the proteoglycan syndecan-1 on the membrane cell surface. The interaction between non-complexed crotamine and syndecan-1 was evaluated previously (Campeiro et al., 2019)
- How many cells taken into the analysis to calculate %positive overlapping staining in Figure 4I and 5M?
Thank you for your observation. 20 cells were evaluated per calculation. This information is added to the Legends of the Figures.
- Are the crotamine/siRNA nanocomplexes biocompatible? At least, a cell cytotoxicity study should be conducted in vitro to evaluate the safety.
We are grateful for the reviewer observation. In fact an extensive evaluation on crotamine cytotoxicity on this specific cell line was performed in Campeiro et al 2018, also evaluating the role of the proteoglycan Syndecan-1 in crotamine internalization and cytotoxicity. The crotamine concentrations used for in vitro studies in this current work (about 300 nM) were below the IC50 indicate in the previous studies (~18 µM).
- Figure 6 well demonstrated that crotamine/siRNA nanocomplexes can accumulate into kidney tissues after 2hr IP injection. Have the authors look at the distribution in other tissues? And what’s the knockdown level of Syndecan-1 in these kidney tissues?
We are grateful for the reviewer remark on this matter. Other tissues were evaluated and no significant accumulation was observed, which is now mentioned in the Results section of the new draft. The knockdown level of Syndecan-1 could be evaluated by measuring the mRNA level on these tissue samples. However, in vivo experiments were performed here with fluorescent scrambled siRNA/crotamine to indicate the biodistribution of the complex. Further experiments using siRNA targeting syndecan-1 will be addressed in a further project, nanocomplex concentration and dosis scheme need to be optimized.
Round 2
Reviewer 1 Report
The authors have addressed most of my comments. However, I have a query regarding the cytotoxicity profile of these formulations at different ratios and concentrations. Have the authors checked the cytotoxicity profiles of these formulations?
Author Response
The authors have addressed most of my comments. However, I have a query regarding the cytotoxicity profile of these formulations at different ratios and concentrations. Have the authors checked the cytotoxicity profiles of these formulations?
We thank the reviewer for bringing up this issue, since this is indeed an important point. Yes, in all FACS experiments we included propidium iodide in order to exclude dead cells. As shown in the Figure below the crotamine:siRNA complexes in 50:1 and 100:1 ratio’s, (siRNA being 3 nM), the cell viability is around 85% and comparable to control cell cultures without incubation by these crotamine:siRNA complexes. This demonstrate the non-toxicity of the used crotamine:siRNA nanoparticles. Of note, the viability after siRNA transfection using lipofectamine is much lower (69%). Please, see the attachment below. We added this info into the Discussion of the new draft of the manuscript.

Reviewer 3 Report
The authors have addressed most of the questions appropriately. Please consider highlighting all revised text and including the page/paragraph/line number when providing the response to the questions.
Author Response
The authors have addressed most of the questions appropriately. Please consider highlighting all revised text and including the page/paragraph/line number when providing the response to the questions.
In the new submission we now included a version of the draft where all changes are indicated, and in this response letter, we now indicate page/paragraph/line numbers below, refering to the clean version of the manuscript.
The manuscript entitled “Crotamine/siRNA nanocomplexes for functional downregulation of syndecan-1 in renal proximal tubular epithelial cells” by Joana D`Arc Campeiro et al. designed and formulated a nanocomplex composed a cell penetration peptide-crotamine and a siRNA targeting to Syndecan-1 gene for proximal tubular epithelia cells (PTECs) delivery and gene regulation in complement pathway. It demonstrated that crotamine/siRNA complex can internalize into PTECs and downregulate Syndecan-1 gene expression efficiently by a series of in vitro functional studies. After IP injection in mouse model, those nanocomplexes were observed to accumulate in kidney tissues by histological staining study. A relative comprehensive investigation on the property and function of crotamine/siRNA nanocomplexes in this manuscript. However, following questions still need to be addressed:
- In Introduction section, the authors should include the current research progress and the barricade for PTECs delivery to give an overview before introducing crotamine as a potential targeting ligand.
We are grateful for the reviewer observation. The topic was clarified in an extra paragraph in the Introduction section (page 4, lines 1-11). Barricades for PTECs delivery, especially the glomerular filtration barrier, are mentioned in the Discussion part of the manuscript (page 17, line 17 to page 18, line 7).
- The detail of design and preparation of crotamine/siRNA is lacking. It’s not clear how the nanocomplexes form, by chemical conjugation or by electrostatic force? The authors mentioned that the complexes are made by vertex and then used immediately at line 77-78. It raises a question that how stable the nanocomplexes are in a regular storage condition. “by vortex” is too general, any speed limit or time requirement? And at line 76, it’s not appropriate to use “(nM)” as the unit for “molar ratio”.
We are very appreciative for the reviewer observation on this matter, the topic was clarified in Method section (page 6, line 12).
- Please include the siRNA sequence in the method part. How is the siRNA sequence selected? Is there any chemical modification on siRNA?
We are very appreciative for the reviewer observation on this matter, the sequence was clarified in the Supplemental Method section (page 1, line 9). Selection was by the company without any siRNA modification.
- In Figure1, the authors attempted to characterize crotamine/siRNA complex by gel electrophoresis. From the gel images, there’s not a uniform and overlapping band of siRNA and crotamine. Are the complexes dissociated under the electrophoresis condition? Figure 1G provides the estimate siRNA complexed rate, but it is not accurate since all the gel bands of siRNA look very smear when complexed with crotamine.
We are grateful for the reviewer remark on this matter. According to Rio DC 2014 (doi:10.1101/pdb.prot080721): “The electrophoretic mobility shift assay (EMSA), or gel mobility shift assay, is a popular and powerful technique for the detection of RNA–protein interactions. It relies on the fact that naked RNA has certain mobility on nondenaturing gels, but if the RNA is bound by protein, the mobility of the RNA is reduced. Therefore, the binding of protein results in a characteristic upward shift of the RNA on a gel.” The complexes did not dissociate under experimental conditions, but stainability of siRNA is progressively reduced under increasing complex formation with crotamine. The crotamine/siRNA interaction is confirmed by the shift occurring in both bands. In Figure 1G the siRNA complexed rate was estimated based on the signal intensity of each siRNA band in Figure 1D when compared to the siRNA band in absence of crotamine (100% intensity, but 0% complexed).
- In Figure2A, it shows that the size of crotamine/siRNA increased over the time. Is it happened before or after adding the salt based stop buffer? And please provide the rationale why the nanocomplex has a smaller size with higher crotamine/siRNA ratio. What’s the PDI of the nanocomplex? Figure 2C indicates that these nanocomplexes are in spherical shape, how can they from sphere? Are these particles homogeneous?
We are grateful for the reviewer remark on this matter. The nanocomplex increasing overtime happens without the salt addition as the presence of salt stops the complex formation by altering the electrostatical interactions. This information was clarified in Methods section (page 6, lines 10-15). Increasing the crotamine/siRNA ratio progressively precludes siRNA molecules to form bridges with crotamine complexed to other siRNA molecules, thereby reducing the formation of larger complexes. PDI of the complexes was always <0,2 as can be seen from the DLS distribution curve. Yes, the complexes were always spherical in form, which apparently is the most favorable form of the siRNA/crotamine complexes formed by electrostatic interactions. Homogeneity of the nanoparticles is shown by distribution of the DLS analysis in 2B and by the small SEM in 2D.
- In Figure 3, the band intensity of lane3 is slightly lower than lane2. Is it caused by partial siRNA degradation? Have the authors studied the serum stability with longer incubation time? And what’s the mechanism that crotamine/siRNA complex can help improve the stability?
We are grateful for the reviewer observation. In fact, the lower intensity in lane 3 might be caused by the partial siRNA degradation, as for this experiment the crotamine/siRNA adopted was 50:1 corresponding to about 75% of siRNA complexed. We did not evaluated the serum stability with longer incubation time as previous studies conducted by the group indicated that for in vitro studies crotamine is internalized by the cells in less than 10 minutes and for in vivo studies in mice, crotamine reaches the kidney in less than 2 hours (Crotamine mediates gene delivery into cells through the binding to heparan sulfate proteoglycans - PubMed (nih.gov), Long term safety of targeted internalization of cell penetrating peptide crotamine into renal proximal tubular epithelial cells in vivo - PubMed (nih.gov)).The siRNA remained intact after exposition of crotamine/siRNA complex to nucleases present in human serum, indicating that the complex structure with crotamine might be preventing the enzymatic degradation of siRNA by nucleases present in serum. We hypothesize that crotamine complexes with siRNA and the excess of crotamine forms a positively charged capsule around the structure protecting the siRNA in the inner part of the nanocomplex structure.
- For cell uptake and colocalization study shown in Figure 4, why the incubation was done at 4 degree rather than 37 degree condition? The endocytosis process is energy driven so the low temperature will slow down the cell uptake in theory.
We are grateful for the reviewer remark on this matter. In fact the incubation at 4 degree was deliberately used to prevent the endocytosis process being able to confirm that the nanocomplex is internalized by this mechanism as it was already shown for other crotamine-complexes (Nascimento et al., 2007; Hayashi et al., 2008; Hayashi et al., 2012). Additionally, the experiment allowed us to evaluate the interaction between the complex and the proteoglycan syndecan-1 on the membrane cell surface. The interaction between non-complexed crotamine and syndecan-1 was evaluated previously (Campeiro et al., 2019)
- How many cells taken into the analysis to calculate %positive overlapping staining in Figure 4I and 5M?
Thank you for your observation. 20 cells were evaluated per calculation. This information is added to the Legends of the Figures (page 28, lines 8-9 and line 23).
- Are the crotamine/siRNA nanocomplexes biocompatible? At least, a cell cytotoxicity study should be conducted in vitro to evaluate the safety.
We are grateful for the reviewer observation. In fact an extensive evaluation on crotamine cytotoxicity on this specific cell line was performed in Campeiro et al 2018, also evaluating the role of the proteoglycan Syndecan-1 in crotamine internalization and cytotoxicity. The crotamine concentrations used for in vitro studies in this current work (about 300 nM) were below the IC50 indicate in the previous studies (~18 µM).
Additional information is given in the attachment and the text below.
We thank the reviewer for bringing up this issue, since this is indeed an important point. Yes, in all FACS experiments we included propidium iodide in order to exclude dead cells. As shown in the Figure below the crotamine:siRNA complexes in 50:1 and 100:1 ratio’s, (siRNA being 3 nM), the cell viability is around 85% and comparable to control cell cultures without incubation by these crotamine:siRNA complexes. This demonstrate the non-toxicity of the used crotamine:siRNA nanoparticles. Of note, the viability after siRNA transfection using lipofectamine is much lower (69%). We added this info into the Discussion of the new draft of the manuscript (page 16, lines 1-2).
